# Biochemical and Antioxidant Profiling of Raspberry Plant Parts for Sustainable Processing

**DOI:** 10.3390/plants12132424

**Published:** 2023-06-23

**Authors:** Audronė Ispiryan, Jonas Viškelis, Pranas Viškelis, Dalia Urbonavičienė, Lina Raudonė

**Affiliations:** 1Lithuanian Research Centre for Agriculture and Forestry, Instituto al. 1, Akademija, LT-58344 Kėdainiai, Lithuania; jonas.viskelis@lammc.lt (J.V.); pranas.viskelis@lammc.lt (P.V.); dalia.urbonaviciene@lammc.lt (D.U.); 2Department of Pharmacognosy, Lithuanian University of Health Sciences, Sukileliu Av. 13, 50162 Kaunas, Lithuania; lina.raudone@lsmu.lt; 3Laboratory of Biopharmaceutical Research, Institute of Pharmaceutical Technologies, Lithuanian University of Health Sciences, Sukileliu Av. 13, 50162 Kaunas, Lithuania

**Keywords:** raspberry morphological parts and characteristics, micronutrients, antioxidant activity, phenolic profile

## Abstract

The optimization of innovation and food technological processes not only increases the profits of companies but also allows them to achieve the set goals of the green trajectory. This research aimed to collect data on the biochemical composition of different parts of the raspberry variety ‘Polka’, including the various morphological parts, to present the importance of differentiating plant parts in food processing, and to show the potential of usage for primary processing in different fields of the food industry. Fruits, stems (cane), leaves, flowers, seeds, and roots were evaluated according to their chemical composition and mineral (Ca, Mg, B, Zn, Cu, Fe, and Mn) contents, phenolic compounds, and antioxidant activity. In our study, the contents of inverted sugar, saccharose, and total sugar varied from 51.8 ± 2.46 %, 18.9 ± 0.31 %, and 69.7 ± 4,36 % in raspberry puree to 5.9 ± %, 1.51 ± %, 7.39 ± % in the seeds, respectively. The results regarding the mineral composition of various raspberry parts (mg/kg) indicated significant differences (*p* < 0.05). The contents of manganese and iron (57.6 ± 0.50; 36.9 ± 0.59) were the highest in all the parts in the plant. Manganese varied from 246 ± 10.32 in inflorescence to 40.1 ± 0.87 in the seeds. Iron fluctuated from 1553 ± 44.03 in the roots to 35.5 ± 0.15 in the seeds. The highest statistically significant boron content (*p* < 0.05) was found in the leaves (41.8 ± 0.33), while the lowest was in the seeds (7.17 ± 0.19). The total phenol content of the raspberry’s distinct parts ranged from 6500 mg GAE/100 g DW to 1700 mg GAE/100 g DW. The inflorescence had the considerably highest total phenol content. Our study found that the highest amount of epicatechin is found in the roots (9162.1 ± 647.86 mg), while the fruits contain only 657.5 ± 92.99, and the lowest value is in the stems (130.3 ± 9.22). High levels of procyanidin B2 were found in the raspberry roots (7268.7 ± 513.98), while the stems had the lowest value–368.4 ± 26.05. The DPPH of the raspberry morphological parts ranged from 145.1 to 653.6 µmol TE/g FW, ABTS—from 1091.8 to 243.4 µmol TE/g FW, and the FRAP—from 720.0 to 127.0 µmol TE/g FW. The study revealed the importance of differentiating plant parts in production for the quality of the final product. Studies showed that raspberry plant parts represent a potential source of natural food ingredients, and can be a potential raw material for products rich in phenolic compounds or dietary fiber, which can provide healthy properties to food when used as an additive that may be economically attractive for consumers.

## 1. Introduction

Food waste and the formation of by-products present a major problem, with adverse environmental, economic, and social impacts. The agri-food industry generates a large proportion of waste and by-products, which account for 40–50% of all emissions from various plant sources, such as peel, pulp, skin, roots, stems, leaves, and seeds [1]. Reducing food waste is a common goal worldwide, and there is a known seriousness about the problem; thus, the EU and many other countries are promoting action plans to reduce food waste, such as the farm-to-table strategy, the circular economy action plan, and EU waste legislation. Reducing food waste could be an important solution for reducing production costs and creating more efficient food systems. In addition, more environmentally sustainable food systems can be developed, and food safety and nutrition can be improved [2].

Berry fruits contain a significant number of diverse bioactive compounds, which individually or in combination can have a positive effect on human health. Therefore, raspberries can be recommended as a natural source of antioxidants. Small fruits are an excellent source of natural antioxidant substances, which is one of the major reasons for their increasing popularity in the human diet. Extracts of fruits from various blackberry, raspberry, and gooseberry cultivars act effectively as free radical inhibitors [3].

Fruits are also abundant in different bioactive compounds, including phytochemicals (phenolic acids, flavonoids, carotenoids, tannins, lignans, and stilbenes), vitamins (provitamin A, C, E, and K), minerals (potassium, calcium, and magnesium), phenolic compound, antioxidant activity and dietary fibers, which play a critical role in human health by alleviating several chronic diseases, mainly coronary heart diseases, cancers, diabetes, cataracts and so on [4,5,6]. Polyphenols are secondary metabolites from plant metabolism, and this category of compounds can be classified into phenolic acids (C6-C1 and C6-C3 skeleton for hydroxybenzoic and hydroxycinnamic acids, respectively), flavonoids (C6-C3-C6 skeleton), stilbenes (C6-C2-C6 skeleton), lignans (C6-C3), and other polyphenols (variable skeleton, such as tyrosol) [7]. As such, raspberries contain vegetative and fructifying organs. Different parts of the raspberry plant have different profiles of bioactive compounds and possible target extraction. For these reasons, they can be differentiated in the food, beverage, and cosmetic industries. Vegetative organs serve to maintain the life of the individual and are differentiated into roots, stems, and leaves. Fructifying organs or reproductive organs (fruits, flowers, and seeds) enable the survival of the species. The biochemical and antioxidant profiling of raspberry fruit and leaves has been widely studied [8,9,10,11]. In addition, globalization has increased the demand for different types of products. Food processing is the set of methods and techniques used to transform raw ingredients into finished and semi-finished products. A significant aspect of food technology is to promote sustainability to avoid waste, save and utilize all the food produced and ensure safe and sustainable processing practices. Professionals in food processing need to be knowledgeable about the general characteristics of raw food materials and the principles of food preservation. Therefore, scientific novelty was revealed by identifying the raw materials formed during the processing of raspberries using different technologies.

Primary processing involves cutting, cleaning, packaging, and the storage, and refrigeration of raw foods to ensure that they are not spoilt before they reach the consumer. These minimally processed foods retain the original properties, i.e., the nutritive, physical, sensory, and chemical properties as in the unprocessed form, and are ready for further processing by the food industry (secondary processing). Secondary food production involves converting raw food ingredients into more useful or edible forms. Secondary food products are refined, purified, extracted, or transformed from minimally processed primary food products.

Some positive impacts of primary food processing are, for example, the increase in shelf life and nutrient bioavailability. However, food processing can also have negative impacts, such as a high content of artificial additives and loss of nutrients. This study shows how different methods of extracting raspberry seed oil, which is what remains after the oil has been extracted, affect the result in creating products based on circular economy principles.

Therefore, the goal of all production is to grow a cultivar with high yield and excellent quality, to create high-quality products, and sell a maximum amount. The study aimed to compare the chemical characteristics as well as the content of antioxidants (anthocyanins, vitamin C, and total phenolics) in different parts of the raspberry plant.

Ponder and Hallman in 2019 tested the content of phenolic compounds in the leaves of raspberry plants of different cultivars (including ‘Polka’) [12]. Their results showed significant differences between the raspberries from organic and conventional systems. Lebedev et al., in 2022, compared the phenolic content of fruits and leaves of raspberry cultivars. Majewski, in 2020, evaluated the phenolic compound content of raspberry seeds [13]. In 2023, Kobori et al. profiled the phenolic compound composition of raspberry flowers [14].

However, all these authors analyzed only one or a few of the plant parts, and there are not enough research studies or published articles evaluating the potential of all the different parts of raspberries and their quality characteristics from one region, the same climatic conditions, and the same soil management and cultivation principles. Therefore, the goal is to highlight and determine the differences in the plant parts of raspberries grown under the same conditions and to evaluate the individual parts of the plant ‘Polka’, the most widely grown raspberry variety in Europe, according to their quality characteristics for waste-free processing.

Potential differences in the biochemical composition and micronutrient capacity of the analyzed raspberry plant parts may be useful in making production decisions in the processing or development of new products and in improving commercial performance accordingly. This means that comparing individual parts of the plant with different nutritional and antioxidant values, in addition to the standard product, can increase the number of products, be competitive, and successfully develop waste-free raspberry processing technologies. It is useful to compare the level of basic compounds from different parts of the plants to underline these differences and to point out the most interesting and promising parts for the food, cosmetic, and pharmaceutical industries. It is important to study all parts of the plant, especially for the determination of phenolic compounds. This research aimed to collect data on the biochemical composition of different morphological parts of the raspberry variety ‘Polka’, to present the importance of differentiating plant parts in food processing, and to show the potential for usage for primary processing in various fields of the food industry.

## 2. Materials and methods

### 2.1. Plant Material and Its Preparation

In the first stage, the primocane fruiting red raspberry cultivar ‘Polka’ was acquired from the raspberry farm, located in North Lithuania (55°47’55.0”N 22°44’57.4”E 55.798603, 22.749268). The average air temperature in Lithuania in August was 16.1 °C. Randomly selected raspberry parts were harvested in August 2021 at physiological maturity (phenological phase 8 (fruit maturity) in the BBCH system) in the morning, and transported to the laboratory of the Institute of Horticulture of the Lithuanian Research Centre for Agriculture and Forestry. Raspberry plant parts (leaves, stems, roots, buds, inflorescence, and fruits) were collected separately and randomly in an area of approximately 50 m^2^. All parts of the raspberry plant in the fields were grouped and immediately taken to the laboratory of the Lithuanian Institute of Agronomy and Forestry, where they were frozen and lyophilized. Raspberry seeds were obtained by separating them using a “Voran” destoning and pulping machine. The seeds were dried at approximately 40 °C. In the second stage, all parts were grounded in a rotary hammer mill SR 300, 200–240 V, 50/60 Hz Retch (Germany) using a 0.5 mm sieve and stored in glass jars until analyses.

### 2.2. Reagents

Analytical and HPLC-grade solvents and reagents were used for chemical analyses. Acetonitrile (99.9%) and methanol (99.9%), potassium persulphate (99%), and 2,2′-azino-bis(3-ethylbenzothiazoline-6-sulfonic acid) diammonium salt (98%) (ABTS), and the reference compounds were obtained from Sigma-Aldrich (Steinheim, Germany); trifluoroacetic acid (≥99%), Trolox (≥98%) and apigenin were supplied from Fluka Chemika (Buchs, Switzerland). The purified deionized water (18.2 mΩ/cm) was produced using the Millipore Simpak1 Synergy 185 ultra-pure (Bedford, MA, USA) water system.

### 2.3. Extraction and Analysis

The powdered samples of approximately 1 g (accuracy 0.0001 g) were extracted with 10 mL of 70% acetone for 15 min in an ultra-sonic bath; 480 W ultrasonic power at 35 kHz was used in the study by using the Sonorex Digital 10 P ultrasonic bath (Bandelin Electronic GmbH & Co. KG, Berlin, Germany). The extracts were filtered through 0.22 µm PDVF syringe filters (Carl Roth GmbH & Co. KG, Berlin, Germany) and stored at 4 °C until analysis. HPLC analysis was performed according to Raudone et al. [15]; the quantitative determination of the compounds identified in the extracts was carried out using five-level calibration graphs of reference compounds for each analyte, and was based on the dependence of the area of the chromatographic peak of the analyte on the concentration of the analyte in the analyzed extract. The amounts of dihydrochalcones, monomeric and oligomeric flavan-3-ols in the tested extracts were calculated at a wavelength of 280 nm, phenolic acids at 320 nm, and flavanols at 360 nm.

### 2.4. Determination of Antioxidant Capacity

An ABTS + radical cation decolorization assay was adjusted according to the methodology described by Re and colleagues, with some modifications. A volume of 3 mL of ABTS + (2,2′-azino-bis(3-ethylbenzthiazoline-6-sulphonic acid)) solution (absorbance 0.800 ± 0.02) was mixed with 20 µL of samples. The absorbance decreasing of each sample was measured at 734 nm in a Cintra 202 (GBC Scientific Equipment, Knox, Braeside, VIC, Australia) spectrophotometer after 30 min. The DPPH• free radical scavenging activity was established using the method suggested by Brand Williams, Cuvelie, and Berset [16], with some modifications: 2 mL DPPH• (2,2-diphenyl-1-picrylhydrazyl) solution in 96.0% *v*/*v* ethanol was mixed with 20 µL of samples. A decrease in absorbance was determined at 515 nm in a Cintra 202 (GBC Scientific Equipment, Knox, Australia) spectrophotometer after 30 min. The ferric-reducing antioxidant power (FRAP) assay was accomplished as described by Benzie and Strain, with some modifications. The FRAP solution consisted of TPTZ (0.01 M dissolved in 0.04 M HCl), FeCl_3_ × 6H_2_O (0.02 M in water), and acetate buffer (0.3 M, pH 3.6) at the ratio of 1:1:10. A volume of 3 mL of a recently prepared FRAP reagent was mixed with 2 µL of samples. The absorbance increase was established at 593 nm in a Cintra 202 (GBC Scientific Equipment, Knox, Australia) spectrophotometer after 30 min. Calculation of all antioxidant activity assays was carried out using Trolox calibration curves, and expressed as µmol of the Trolox equivalent (TE) per one gram of dry weight (µmol TE/g DW) [17].

### 2.5. Determination of Titratable Acidity

A portion of the premixed sample is taken and filtered through cotton wool, filter paper, or cloth. Transfer 25 mL of the filtrate to a 250 mL volumetric flask, dilute to the mark with water, and mix well. Depending on the expected acidity, add 25 mL, 50 mL, or 100 mL of the diluted sample to a conical flask, add 3–5 drops of phenolphthalein, and titrate with 0.1 M sodium hydroxide solution.

### 2.6. Determination of Ascorbic Acid (Vitamin C) Content

A total of 20 mL of 1% HCl is added to 10 g of the test substance in a mortar and quickly crushed to a homogeneous mass. The resulting mass is poured through a funnel into a measuring flask with a capacity of 100 mL. The mortar is washed with a 1% oxalic acid solution, pouring the washing solution into the same measuring flask. The contents of the flask are diluted to the mark with a 1% oxalic acid solution. The flask is closed with a cork, shaken, and left to stand for 5 min. The mixture in the flask is filtered through a dry filter into a dry flask. Two portions of 10 mL each are taken from the filtrate and poured into 50 mL flasks. Titrate with micro burettes with 0.001 N 2,6-dichlorophenolindophenol solution until a bright pink color does not disappear for 0.5–1 min.

### 2.7. Determination of Sugars

Monosaccharides, sucrose, and total sugar content in samples were determined by the Bertrand method, which is based on the reducing action of sugar on the alkaline solution of tartrate complex with cupric ion; the cuprous oxide formed is dissolved in a warm acid solution of ferric alum. The ferric alum is reduced to FeSO_4_ which is titrated against standardized KMnO_4_; Cu equivalence is correlated with the table to obtain the amount of reducing sugar.

### 2.8. Determination of Dry Matter

Dry matter content was determined gravimetrically by drying apple samples to a constant weight at 105 °C.

### 2.9. Determination of the Amount of Macro-and Microelements

The amount of macro- and microelements in the raspberries was determined by the spectrometric method. Mineralization of dry raw material was carried out with a microwave mineralizer Multiwave GO (Anton Paar High-precision Instruments, Austria). A total of 0.5 g of dry raw material was poured into 5 mL of 65% nitric acid and 3 mL of 30% hydrogen peroxide. The samples were mineralized in several stages, using time and temperature regimes: 1st stage—a temperature of 150 °C was reached in 3 min and held for 10 min; 2nd stage—a temperature of 180 °C was reached within 10 min and held for 10 min. After the mineralization steps, the sample was diluted to 50 mL with deionized water. The composition and quantity of macro- and microelements were measured using an inductively coupled plasma optical emission spectrometer (ICP-OEC) SPECTRO GENESIS (Spectro Analytical Instruments, Germany). ICP-OEC parameters: power 1300 W, plasma flow rate 12 L min^−1^, make-up flow rate 1 L min^−1^, nozzle flow rate 0.8 L min^−1^, and sample aspiration rate 1 mL min^−1^. Identification wavelengths of individual elements: B (249.773 nm), Ca (445.478 nm), Cu (324.754 nm), Fe (259.941 nm), K (766.491 nm), Mg (279.079 nm), Mn (259.373), Na (589.592 nm), P (213.618), S (182.034 nm) and Zn (213.856 nm). Calibration standards were prepared by diluting the multi-element standard solution (1000 mg L^−1^, Merck, Germany) with 6.5% nitric acid. Sulfur and phosphorus standard solutions (1000 mg L^−1^, Merck, Germany) were diluted with deionized water. The amount of macronutrients was expressed in mg g^−1^, and micronutrients in µg g^−1^ dry weight.

Soil studies were conducted in the “Agrochemical Research Laboratory” accredited by the LAMMC Institute of Agriculture. In the spring, the combination of 500 g of soil samples from the arable layer (0–20 cm deep) was taken from each option in the repetition box (from 5 compartments). Soil samples in the laboratory were dried to the mass of the air, crushed in a porcelain mortar, and sifted through a 2 mm sieve.

### 2.10. Determination of Soil Properties

Soil studies were conducted in the “Agrochemical Research Laboratory” accredited by the LAMMC Institute of Agriculture. In the summer, the combination of 1000 g of soil samples from the arable layer (0–20 cm deep) was taken from each option in the repetition box (from 10 compartments). Soil samples in the laboratory were dried to the mass of the air, crushed in a porcelain mortar, and sifted through a 1 mm sieve.

The following soil agrochemical parameters are estimated:Humus content (%) is determined by the amount of organic and general carbon in the sample after dry burning (ISO 10694: 1995).Soil reaction pH-potentiometric method, pH-meter 1n KCl excerpt (LST ISO 10390: 2005).The amount of mineral nitrogen (MG KG -1) is calculated by adding nitrate, nitrite, and ammonia nitrogen (ISO 14265-2: 2005).The amount of mobile phosphorus (P_2_O_5_) and mobile potassium (K_2_O) is calculated using the Egner–Rimo–Domingo (A–L) method [18].

### 2.11. Statistical Analysis

All the experiments were carried out in triplicate. In the statistical processing of the data obtained from the analysis of the chemical composition of the fruits, the standard deviation was calculated and presented together with the mean values. MS Excel (USA) and IBM SPSS Statistics (USA) software packages were used for statistical analysis. One-way analysis of variance (ANOVA), along with the post hoc Tukey’s HSD test, was employed for statistical analysis. Differences were considered to be significant at *p* < 0.05. The antioxidant activity was evaluated by using ABTS, DPPH, and FRAP assays. Values were expressed as means with standard deviation error bars.

## 3. Results and Discussion

T. Turmanidze et al. [19] and E. Carvalho et al. [20] reported that whole berry extracts usually contain significant amounts of ascorbic acid and carotenoids. According to our data, the ascorbic acid contents ranged from 0.063 to 0.147 g/100 g DW, with raspberry puree holding the highest content and roots having the lowest content. In a study of 24 different raspberry cultivars, Yu et al. found that the contents of total sugar and reducing sugar varied from 50.78 ± 1.99 to 82.64 ± 0.21 g/100 g DW, and 30.26 ± 0.33 to 55.76 ± 2.66 g/100 g DW, respectively [21]. In our study, contents of inverted sugar, saccharose, and total sugar varied from 51.8 ± 2.46, %, 18.9 ± 0.31, %, and 69.7 ± 4.36, % in the raspberry puree to 5.9 ± %, 1.51 ± %, 7.39 ± % in the seeds, respectively. The values of dry matter varied from 98.4 ± 3.55 in the stems and 98.4 ± 4.44 in the leaves to 83.8 ± 3.83 in the raspberry fruits.

The values of titratable acidity (TAC) varied from 20.9 ± 0.32 a, %, in the unripe raspberries to 2.07 ± 0.06 f. Comparing raspberries with seeds, raspberry puree, and non-billed raspberry acidity, there were significant differences in total acidity, which is extremely important in fermented drinks and wine production. The TAC values were higher than most of the data determined by others [22], where the values of TAC varied from 4.90 ± 1.19 to 17.51 ± 0.51 g/100 g DW. This might be due to the strong relationship between the berry’s acidity and climate conditions. In August 2021, when the raw materials were collected for investigations, unusually high rainfall (127 mm/month) was collected, while in 2020 precipitation was only 46.7 mm/month in the same month (data obtained from Šiauliai Meteorological Station). According to the meteorological data obtained, it can be stated that the year 2021 was less favorable for raspberry development and growth, which may have created stressful conditions for plants and encouraged larger amounts of some secondary metabolites. The inverted sugar, saccharose, total sugar, ascorbic acid (Vitamin C), titratable acidity, and dry matter are shown in Table 1.

Product characteristics such as sweetness, acidity, and juiciness are important for consumers, so it is necessary to take this into account when processing fruit. Quality is influenced by the amount and composition of sugar accumulated in the fruit. Sweet and sour are produced by sugar and acid, respectively. Their contributions to the taste depend not only on the levels of sugar and acid but also on the types and relative proportions of sugar and acid. Therefore, it is very important to determine the composition and amounts of sugar and acids in berries, and for improving the quality of raspberry production it is worth conducting experiments that would help determine the amounts of additional chemicals, preservatives, and food additives, because they can stimulate their production and better meet the needs of the consumer.

Total sugars include all sugars, whatever their food source (whether added or naturally present in foods), i.e., all monosaccharides and disaccharides. The amount of total sugars is provided in nutrition labeling in the EU. Regulation (EU) No 1169/2011 on the provision of food information to consumers harmonizes how sugars must be labeled. The nutrition declaration must indicate the amount of total sugars, and in the ingredients list the types of sugars added must be declared.

Sugars play a key role in different foodstuffs. As well as bringing sweetness, they also have important biological, sensory, physical, and chemical properties. For example, sugars help provide the taste, texture, and color of foods and extend their shelf-life, which preserves the safety and quality of the food. Sugars can in some cases be reduced/replaced, but no other single ingredient can replace all the functions of sugars.

An important indicator of raspberry quality is the chemical composition, especially the sugar content, and the acidity in wine production; the soluble dry matter is important in the production for fruit respiration and weight loss. Sugars and acidity are also important indicators determining the organoleptic properties of fruits [23,24,25].

### 3.1. Mineral Composition of Raspberry Parts

An adequate ratio of micronutrients and their favorable content in the soil, whose uptake can be affected by low soil pH, ensures the plant’s optimal supply [26]. The chemical analysis of an average soil sample showed that the soil had a slightly acid reaction (pH 5.6), a high level of humus (6.1%), a moderate content of available phosphorus (136 mg/kg) and a high content of available potassium (167 mg/kg) [27]. Soil properties in the raspberry growing area are presented in Table 2 below.

The results regarding the mineral composition of different raspberry parts (mg/kg) indicated significant differences (*p* < 0.05). The contents of manganese and iron (57.6 ± 0.50; 36.9 ± 0.59) were the highest in all parts of the plant; manganese varied from 246 ± 10.32 in inflorescence to 40.1 ± 0.87 in the seeds, and iron varied from 1553 ± 44.03 in the roots to 35.5 ± 0.15 in the seeds. The highest statistically significant boron content (*p* < 0.05) was found in the leaves (41.8 ± 0.33), while the lowest was in the seeds (7.17 ± 0.19). The lowest contents, i.e., Ca (0.13 mg), Mg (0.11 mg), B (7.23 mg), Cu (2.93 mg), and Fe (35.6 mg) were found in the seeds. The chemical composition of the micronutrients of different morphological parts of the raspberry is presented in Table 3.

The minimum values were for Ca and Mg (from 0.13 in the seeds to 1.15 in the leaves for calcium and from 0.11 in the seeds to 0.47 in the leaves for magnesium). Manganese and iron were mainly found (manganese content was from 40.4 mg/kg in the seeds to 248.7 mg/kg in the flowers. Iron content was from 35.6 mg/kg in the seeds to 1551 mg/kg in the roots). High iron content can be explained by the iron content in the environment, since raspberries accumulate large amounts of iron from the environment. However, in 2022 the Sikirić et al. study showed that no significant correlation was found between the content of Fe in the soil and plant organs, or between the Fe in leaves and fruits [28].

The copper contents determined by the above-mentioned Karaklajić-Stajić et al. were slightly lower than our results for the leaves. Cu content in raspberry leaves from Serbia ranged from 3.00 ± 0.07 to 4.00 ± 0.08 μg/g, compared to ours of 4.96 ± 0.30 d. The highest result for copper was found in the unripe raspberries—10.3 ± 0.45 a.

Dresler et al. [29], who studied raspberry leaves, found that, depending on the region of cultivation, the mean content of Mg in the raspberries ranged from 0.26% to 0.45%, compared to our 0.47%, and the mean Ca content ranged from 0.72% to 1.55%, compared to our 1.15%. The raspberry leaves were characterized by a very high content of Mn; the mean concentration of this element was 702 mg kg^−1^, compared to our 166 mg kg^−1^. The mean Fe concentration in the plants was 191 mg kg^−1^, compared to our 81.8 mg kg^−1^. The B concentration in the raspberry leaves ranged from 25.5 to 128.5 mg kg^−1^, compared to our 41.8 mg kg^−1^. The mean Zn concentration in the raspberry leaves was 40.3 mg kg^−1^, while our results for Mg were 0.47%.

### 3.2. The Total Phenolic Content

Phenolic compounds are the most common secondary metabolites of plants [30]. Plant extracts and phenolic compounds exert protective effects against oxidative stress and inflammation caused by airborne particulate matter, in addition to a range of anti-inflammatory, anticancer, anti-aging, antibacterial, and antiviral activities [31]. Phenolic acids, readily absorbed through intestinal tract walls, are beneficial to human health, due to their potential antioxidants, and avert the damage of cells resulting from free-radical oxidation reactions. On regular eating, phenolic acids also promote the anti-inflammation capacity of human beings.

The total phenol content of the raspberry’s different parts ranged from 6500 mg GAE/100 g DW to 1700 mg GAE/100 g DW (Table 1 and Figure 1). The inflorescence had a considerably highest total phenol content. The high value was also detected in berries, and was 26% higher than that detected for the lowest value in the seeds. The amount of ascorbic acid content ranged from 60 to 140 mg/100 g of dry weight. The concentration of total phenol quantified in raspberry parts was higher than other scientists have determined [32]. These differences could be due to cultivation principles, environmental characteristics, and soil characteristics.

In addition to exploring the potential protective effects, phenolic compounds provide health benefits against chronic diseases, considering the modifications during food processing techniques, and therefore, overall bioavailability is essential. Plant extracts and phenolic compounds exert protective effects against oxidative stress and inflammation caused by airborne particulate matter, in addition to a range of anti-inflammatory, anticancer, anti-aging, antibacterial, and antiviral activities [33]. Therefore, phenolic compounds can be used in the pharmaceutical industry as therapeutic agents. The antioxidant and antimicrobial properties enable phenolic compounds to function as food preservatives and additives. Thus, they also have applications in the food industry. Innovations in technology and production force countries to compete for high-value products. The distribution of raspberry plant parts and the conducted research help to optimize production processes, to use berry pomace and to identify sustainable valorization processing methods.

### 3.3. Individual Phenolic Profile

Kaempferol-3-beta-O-glucuronide is a flavonoid glucuronide, which can be found in plants and is deconjugated by microsomal β-glucuronidase from various human cells. It has a role as a metabolite. It is a kaempferol O-glucuronide and a trihydroxyflavone. Kaempferol shows a wide range of pharmacological activities, including anti-inflammatory and antioxidant effects, has a liver-protecting effect, and may be associated with a reduced risk of developing Alzheimer’s disease. This natural compound also has great pharmacological capability, and is now considered to be an alternative cancer treatment [34].

The highest statistically significant amount of kaempferol O-glucuronide was found in leaves (4088.6 ± 289.11 a). However, this element was found very little in other parts of the plant, for example, the lowest value—only 4.0 ± 0.29 b—was found in the stems, so to extract this element, the leaves should be separated from the stems in production.

Epicatechin is one of the most investigated catechins, due to its diverse biological properties [35]. People who regularly consume a plant-based diet will have a good amount of epicatechin circulating in their blood. These compounds have demonstrated diverse biological functions such as anti-proliferative, anti-inflammatory, antioxidant, antimicrobial, and cardio-protective activities. The major biological properties of epicatechin are studied using both in vitro and in vivo models. In vitro studies suggested that the antioxidant activity of epicatechin is mainly due to its ability to scavenge free radicals through the multiple phenolic groups attached. Similarly, epicatechins also showed significant antimicrobial activity against various multidrug-resistant pathogens, which is a serious need of today’s healthcare system; epicatechin also increases the capacity for muscle aerobic metabolism, thereby delaying the onset of fatigue [36].

In the scientific literature, the authors indicate that epicatechin is found in smaller concentrations in berries and most of the regularly consumed fruits, chocolates, and non-alcoholic beverages. However, our study found that the highest content of epicatechin is found in the roots (9162.1 ± 647.86 mg), while the fruits contain only 657.5 ± 92.99, and the lowest value is in the stems (130.3 ± 9.22). The highest content of epigallocatechin was determined in the inflorescence (5882.6 ± 415.96) and leaves (3444.3 ± 243.55 b). The lowest values of epigallocatechin were in the stems (449.2 ± 31.76) and roots (542.9 ± 38.39).

High levels of procyanidin B2 have been found in raspberry roots (7268.7 ± 513.98), while the stems had the lowest value—368.4 ± 26.05. The highest statistically significant amount of chlorogenic acid was found in stems and leaves (3017.6 ± 213.38 and 2154.8 ± 152.37, respectively), while the seeds had only (7.2 ± 0.51). The obtained research results are significant because they reveal what is especially important to know in food and pharmaceutical production, i.e., which part of the plant contains the most, to highlight and offer a product that meets the consumer’s expectations (Table 4).

### 3.4. Antioxidant Activity

The antioxidant activity (AA) of raspberry parts was evaluated using DPPH, ABTS, and FRAP assays. It was observed that different raspberry parts vary significantly in the quantity of antioxidant activity. The DPPH of raspberry morphological parts ranged from 145.1 to 653.6 µmol TE/g FW, and ABTS—from 1091.8 to 243.4 µmol TE/g FW— and the FRAP—from 720.0 to 127.0 µmol TE/g FW. According to all assays, inflorescence showed the highest antioxidant activity from 1091.8 µmol TE/g FW using ABTS to 653.6 µmol TE/g FW using DPPH. The lowest AA was in the seeds (DPPH—145.1 µmol TE/g FW and FRAP—127.0). It was established that different morphological parts of the raspberry plant have statistically significantly different antioxidant activity.

The research results in the Table 5 and Figure 2 demonstrate the high antioxidant activity of raspberry inflorescence. This should encourage food and pharmaceutical manufacturers to create products from them in response to consumer demand for such products. Raspberry roots are also a very good source of natural antioxidants, and can be alluded to as “superfoods” or “functional foods”.

## 4. Conclusions

In conclusion, this research shows that different morphological parts of the raspberry plant represent a potential source of natural food ingredients. To extract certain elements from the plant, it is necessary to find out which part of it has most of the required elements. Dissecting the different morphological parts of the plant in production would give them added value (nutritional and functional) and obtain higher productivity and higher quality production. The first study comparing the chemical composition of individual parts of the different morphological parts of the raspberry is particularly significant in the development of waste-free technologies, increasing the economic value of raspberry farms.

More and more people are analyzing product label information and paying close attention to ingredients. Regulation (EU) of the European Parliament and Council outline the requirements established by 1169/2011, which apply to food sold throughout the European Union, according to which the food manufacturer must provide all information about the composition or nutritional content of the product. Food labeling rules are created to inform and protect the consumer, because it is the information provided on the food product package, on the label attached to it, or next to the food product that helps the consumer to evaluate and choose the right food product. Voluntarily provided information about the product draws the attention of consumers and encourages consumer choice. It should be noted that not only technical parameters are regulated, but also claims about nutritional properties and health benefits of products, which must be scientifically proven, and manufacturers or importers must be able to provide scientific documents supporting this. Therefore, from a future perspective, this study can help the manufacturer to inform the consumer in more detail when providing information on vitamins and minerals under Annex XIII of the regulation, and supplement scientific research by conducting a consumer needs study according to the above-mentioned regulation. Our study revealed the importance of differentiating plant parts in production for the quality of the final product. For example, by removing the seeds or separating the leaves from the stems, products with a completely different chemical composition can be obtained. This can be highlighted by providing information to the end user in the product labeling.

The data provided by the study confirm the need to properly optimize the processing of raspberries by exploiting all parts of the plant according to its biochemical compounds, to strengthen the marketing of the products sold through areas such as labeling, consumer information, and presentation of the actual composition of the product. Studies revealed that raspberry plant parts represent a potential source of natural food ingredients and can be considered as a potential raw material for products rich in phenolic compounds or dietary fiber, which can provide healthy properties to food when used as an additive that may be economically attractive for consumers. In the future, it would be appropriate to study the processing technologies of plant parts such as flowers or roots, as they have been little researched.

## Figures and Tables

**Figure 1 plants-12-02424-f001:**
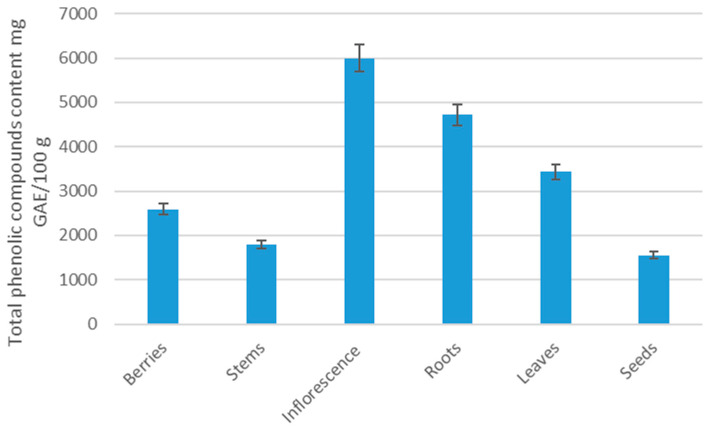
Total phenolic content in the samples of morphological parts of the raspberry plant, mg GRE/100 g DW.

**Figure 2 plants-12-02424-f002:**
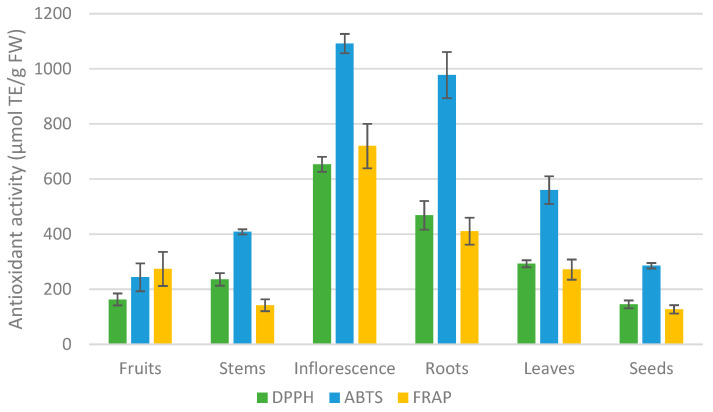
The antioxidant activity of the raspberry plant’s different morphological parts was evaluated by using ABTS, DPPH, and FRAP assays. Values were expressed as means with standard deviation error bars.

**Table 1 plants-12-02424-t001:** Quality indexes of different dried raspberry plant parts.

Sample	Inverted Sugar %	Saccharose %	Total Sugar %	Vitamin C mg %	Titratable Acidity %	Dry Matter %
**Raspberry with seeds**	51.7 ± 0.56 a	13.4 ± 0.58 b	63.9 ± 2.85 b	140.0 ± 3.45 a	15.4 ± 0.58 c	83.8 ± 3.83 c
**Raspberry puree**	51.8 ± 2.46 a	18.9 ± 0.31 a	69.7 ± 4.36 a	147.0 ± 6.35 a	20.9 ± 0.32 a	86.1 ± 3.17 bc
**Unripe raspberry**	15.5 ± 0.53 b	1.9 ± 0.11 e	17.1 ± 0.77 c	117.0 ± 3.99 b	17.4 ± 0.81 b	95.0 ± 5.74 abc
**Inflorescence**	14.4 ± 0.45 b	1.25 ± 0.05 e	16.0 ± 0.5 c	95.0 ± 3.57 c	13.2 ± 0.36 d	93.3 ± 2.99 abc
**Leaves**	11.2 ± 0.64 c	3.36 ± 0.05 d	14.7 ± 0.55 c	82.0 ± 2.48 d	8.4 ± 0.25 e	98.4 ± 4.44 a
**Seeds**	5.9 ± 0.23 d	1.5 ± 0.03 e	7.4 ± 0.33 d	720 ± 4.21 de	2.6 ± 0.09 f	96.6 ± 3.84 ab
**Stems**	10.3 ± 0.59 c	5.6 ± 0.15 c	15.9 ± 0.23 c	93.0 ± 0.94 c	3.0 ± 0.17 f	98.4 ± 3.55 a
**Roots**	3.7 ± 0.15 d	1.8 ± 0.05 e	5.6 ± 0.17 d	62.0 ± 1.93 e	2.1 ± 0.06 f	96.2 ± 4.55 ab

Note: data are expressed as average value ± standard deviation of three replicates and the different letters in each column indicate significant differences (*p* < 0.05).

**Table 2 plants-12-02424-t002:** Soil properties in the raspberry growing area.

Soil Properties	Description
Soil location	55°47′42.2″ N 22°44′59.0″ E
Granulometric composition	Loam
pH 1 mol/l KCl in suspension	5.6 ± 0.2
The concentration of mobile phosphorus (P_2_O_2_), mg/kg	136 ± 14
The concentration of mobile potassium (K_2_O), mg/kg	167 ± 11
Nitrogen (sum of nitrate plus nitrite), mg/kg	109.05 ± 7.90
The concentration of nitrogen (ammonia), mg/kg	5.67 ± 1.04
The concentration of mineral nitrogen, mg/kg	114.72 ± 4.66
Organic carbon concentration %	3.89 ± 0.43
Humus concentration	6.71 ± 0.74

Note: data are expressed as average value ± standard deviation of three replicates.

**Table 3 plants-12-02424-t003:** Micronutrient content (% or mg/kg) of different raspberry morphological parts.

Research Parameter	Sample Name and Test Results						
In Dry Matter	Ripe Berries	Unripe Berries	Leaves	Stems	Inflorescence	Seeds	Roots
**Calcium (Ca), %**	0.13 ± 0.00 e	0.59 ± 0.03 d	1.15 ± 0.06 a	0.64 ± 0.03 d	0.73 ± 0.03 c	0.13 ± 0.00 e	0.82 ± 0.02 b
**Magnesium (Mg), %**	0.14 ± 0.01 d	0.31 ± 0.01 c	0.47 ± 0.01 a	0.15 ± 0.01 d	0.37 ± 0.01 b	0.11 ± 0.00 e	0.16 ± 0.03 d
**Boron (B), mg/kg**	17.40 ± 0.49 e	29.00 ± 0.98 c	41.80 ± 0.33 a	22.00 ± 0.10 d	31.70 ± 1.03 b	7.17 ± 0.19 f	17.57 ± 0.35 e
**Zinc (Zn), mg/kg**	21.10 ± 0.77 c	26.10 ± 0.33 b	17.90 ± 1.08 d	17.10 ± 0.70 d	38.50 ± 1.26 a	18.10 ± 0.95 d	18.87 ± 0.25 cd
**Copper (Cu), mg/kg**	6.79 ± 0.31 b	10.30 ± 0.45 a	4.96 ± 0.30 d	3.20 ± 0.15 e	6.04 ± 0.15 c	2.90 ± 0.10 e	6.06 ± 0.06 c
**Iron (Fe), mg/kg**	36.90 ± 0.59 e	57.50 ± 2.76 c	81.80 ± 2.73 b	45.90 ± 0.30 d	59.20 ± 2.27 c	35.50 ± 0.15 e	1553.00 ± 44.03 a
**Manganese (Mn), mg/kg**	57.60 ± 0.50 e	178.00 ± 7.29 b	166.00 ± 8.84 b	119.00 ± 6.79 c	246.00 ± 10.32 a	40.10 ± 0.87 e	76.4 0± 0.44 d

Note: data are expressed as average value ± standard deviation of three replicates and the different letters in each column indicate significant differences (*p* < 0.05).

**Table 4 plants-12-02424-t004:** Phenolic compound profile (µg/g) in different morphological parts of the raspberry plant.

Part of the Berry	Leaves	Stems	Roots	Buds	Inflorescence	Seeds	Berries
**Caffeic acid**	110.2 ± 7.79 a	13.1 ± 0.92 d	0.3 ± 0.02 e	48.6 ± 3.43 c	82.9 ± 5.86 b	2.7 ± 0.19 de	5.9 ± 0.84 de
**Catechin**	461.9 ± 32.67 a	104.1 ± 7.36 d	8.3 ± 0.59 e	28.5 ± 2.02 de	270.5 ± 19.12 c	363.4 ± 25.7 b	525.9 ± 74.37 a
**Chlorogenic acid**	2154.8 ± 152.37 b	3017.6 ± 213.38 a	15.9 ± 1.13 c	129.1 ± 9.13 c	171.0 ± 12.09 c	7.2 ± 0.51 c	15.2 ± 2.15 c
**Epicatechin**	1108.2 ± 78.36 bc	130.3 ± 9.22 d	9162.1 ± 647.86 a	1380.1 ± 97.59 b	606.4 ± 42.88 cd	432.9 ± 30.61 cd	657.5 ± 92.99 cd
**Isoquercitrin**	222.3 ± 15.72 a	31.3 ± 2.22 c	N.D.	39.1 ± 2.77 c	73.4 ± 5.19 b	N.D.	5.8 ± 0.82 d
**Procyanidin A1**	22.9 ± 1.62 d	1.5 ± 0.11 d	134.3 ± 9.49 b	198.8 ± 14.06 a	86.5 ± 6.12 c	202.4 ± 14.31 a	91.6 ± 12.95 c
**Procyanidin B1**	130.5 ± 9.23 a	122.8 ± 8.68 a	14.8 ± 1.05 c	21.2 ± 1.5 c	12.1 ± 0.86 c	94.1 ± 6.65 b	19.1 ± 2.7 c
**Procyanidin B2**	1079.3 ± 76.32 d	368.4 ± 26.05 e	7268.7 ± 513.98 a	4174.4 ± 295.18 b	1806.5 ± 127.74 c	780.6 ± 55.2 de	781.8 ± 110.57 de
**Procyanidin C1**	1145.9 ± 81.03 a	86.5 ± 6.12 d	13.6 ± 0.96 d	557.4 ± 39.41 c	853.9 ± 60.38 b	60.5 ± 4.28 d	487.4 ± 68.92 c
**Quercetin**	27.4 ± 1.94 a	2.8 ± 0.2 d	4.1 ± 0.29 cd	6.0 ± 0.42 c	4.9 ± 0.35 cd	10.0 ± 0.71 b	4.1 ± 0.58 cd
**Salicylic acid**	28.1 ± 1.99 b	8.9 ± 0.63 c	N.D.	3.9 ± 0.28 d	36.6 ± 2.59 a	N.D.	N.D.
**Tiliroside**	533.7 ± 37.74 a	1.3 ± 0.09 c	8.6 ± 0.61 c	14.4 ± 1.02 c	59.9 ± 4.2 b	4.8 ± 0.34 c	0.6 ± 0.08 c
**Kaempferol-3-O-** **glucuromide**	4088.6 ± 289.11 a	4.0 ± 0.29 b	12.9 ± 0.91 b	114.3 ± 8.08 b	267.6 ± 18.92 b	3.4 ± 0.24 b	10.6 ± 1.49 b
**Ellagic acid**	176.3 ± 12.47 b	N.D.	48.5 ± 3.43 d	123.9 ± 8.76 c	439.9 ± 31.1 a	201.7 ± 14.26 b	115.6 ± 16.35 c
**Astragalin**	67.2 ± 4.75 a	10.2 ± 0.72 c	0.1 ± 0.01 d	4.7 ± 0.33 d	20.8 ± 147 b	0.3 ± 0.02 d	0.6 ± 0.08 d
**Epigallocatechin**	3444.3 ± 243.55 b	449.2 ± 31.76 e	542.9 ± 38.39 de	2681.6 ± 189.62 c	5882.6 ± 415.96 a	642.9 ± 45.46 de	1031.3 ± 145.85 d
**Epigallocatechin** **gallate**	408.4 ± 28.88 b	130.1 ± 9.2 d	925.5 ± 65.44 a	287.6 ± 20.34 c	496.9 ± 35.14 b	479.8 ± 33.93 b	174.8 ± 24.71 d

Note: data are expressed as average value ± standard deviation of three replicates and the different letters in each line indicate significant differences (*p* < 0.05).

**Table 5 plants-12-02424-t005:** Antioxidant activity (µmol TE/g FW) of raspberry morphological parts.

Sample	DPPH	ABTS	FRAP
**Fruits**	163.1 ± 21.7 e	243.4 ± 50.5 d	273.9 ± 61.9 c
**Stems**	235.8 ± 22.9 d	408.5 ± 9.5 c	142.2 ± 21.7 d
**Inflorescence**	653.6 ± 27.3 a	1091.8 ± 35.0 a	720.0 ± 80.6 a
**Roots**	468.7 ± 52.1 b	977.6 ± 83.5 a	410.8 ± 48.8 b
**Leaves**	292.7 ± 12.6 c	559.8 ± 50.0 b	271.5 ± 36.8 c
**Seeds**	145.1 ± 14.3 e	285.5 ± 9.9 d	127.0 ± 15.1 d

Note: data are expressed as average value ± standard deviation of three replicates and the different letters in each column indicate significant differences (*p* < 0.05).

## Data Availability

The data presented in the present study are available in the article.

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
