# Peer review of "Biochemical and Antioxidant Profiling of Raspberry Plant Parts for Sustainable Processing"

_plants, 2023, doi:10.3390/plants12132424_

Round 1
Reviewer 1 Report
The paper "Biochemical and antioxidant profiling of raspberry plant parts 2 for sustainable processing" deals with the evaluation of the phenol, sugar, and micronutrient composition of different part of the raspberry plant. The manuscript has poor methodology, scarce discussion of the results, and an unclear aim. For these reasons, I believe that the paper is unsuitable for publication. Some comments for clarifying the evaluation:
The introduction section is not focused on a single issue and the scope of the paper is not clear. It is long and relatively well-written, but is not clear the direction it gets until the aim of the paper is clearly expressed. I believe it should be shortened, and more previous literature shall be mentioned.
Based on their structure, phenolic acids and other small phenolics (e.g., tyrosol) are not technically polyphenols (only one phenol ring). I believe that, throughout the manuscript, "phenolic compounds" would be more appropriate than "polyphenols".
The sentence in rows 87-88 is misleading. It is not true that other parts of raspberry plants other than the fruit have not been studied for their phenolic composition. For example, Ponder and Hallman in 2019 have tested the content of phenolic compounds in leave of raspberry plants of different cultivars (including Polka) (10.3390/antiox8100458). This paper should be been cited in the submitted manuscript. Similarly, Lebedev et al. in 2022 have compared the phenolic content of fruits and leaves of raspberry cultivars (again including Polka) (10.3390/antiox11101961). This paper is even mentioned in the submitted manuscript, contradicting the previous statement. Majewski, in 2020, has evaluated the phenolic compounds content of raspberry seeds (paper not cited, 10.3390/nu12061630). In 2023, Kobori et al. have profiled the phenolic compound composition of raspberry flowers (paper not cited, 10.3390/nutraceuticals3020015). A more detailed research of previously published papers should be carried out.
The methodology, and especially the sample analysis, looks, at best, odd. In tables 1, and 3-5 different samples are reported. 8 distinct samples were analyzed for sugars (table 1), 7 samples (not the same 7 though) were analyzed for micronutrients and individual phenolics (table 3 and 4, respectively), and 6 samples were analyzed for antioxidant activity. Ideally, one would analyze different compounds from the same samples to get an overall ideas of the trends.
The authors should have also explained the extremely surprising results obtained for the different parts of the plant, with berries having among the lowest amounts of most analyzed phenolics. Moreover, despite their supposed extremely rich content in anthocyanins, none of such compounds was included in the analysis.
The manuscript needs to undergo a deep revision of the English language (e.g., rows 54-55) as well as the a check on punctuations (e.g., row 41, 42, 45, 46) and typos (e.g., row 29).
Author Response
Thank You for the constructive and beneficial comments. The all suggestions were included in the article.
On the different research samples: we note that the article was interdisciplinary, and we wanted to provide insights from the practical side for sustainable processing. In Table 1, unripe raspberries were added, because they are used mostly in food production (juices, wines) and these studied parameters are important precisely in this part, while phenolic compounds were studied (Table 4) additionally for buds that can be used in pharmaceuticals.
Ponder and Hallman in 2019 have tested the content of phenolic compounds in leave of raspberry plants of different cultivars (including ‘Polka’). Their results showed significant differences between the raspberries from organic and conventional system. Lebedev et al. in 2022 have compared the phenolic content of fruits and leaves of raspberry cultivars. Majewski, in 2020, has evaluated the phenolic compounds content of raspberry seeds. In 2023, Kobori et al. have profiled the phenolic compound composition of raspberry flowers.
However, all these authors analyzed only one or a few of the plant parts and there are not enough research or published articles evaluating the potential of all different parts of raspberries and their quality characteristics from one region, the same climatic conditions, the same soil management, and cultivation principles. Therefore, the goal is to highlight and determine the differences in plant parts of raspberries grown under the same conditions, evaluate the individual parts of the plant 'Polka', the most widely grown raspberry variety in Europe, according to their quality characteristics for waste-free processing.
The revision in all aspects was done in the article, with the consultation of native English speakers and all co-authors.
Reviewer 2 Report
The study deals with the Biochemical and antioxidant profiling of raspberry plant parts for sustainable processing. Berry fruits contain a significant amount of diverse bioactive compounds, which individually or in combination can have a positive effect on human health. The study is well prepared and present useful information. Therefore, should be consider for publication in this journal. However, there are some decencies which must be improved.
The abstract is not informative.
Methods description and results are not well presented in the abstract section.
Present main findings of the study in the abstract.
Line 44-48 should be cited with recent studies. It lacks references. The following studies could be cited. https://doi.org/10.3390/molecules28083403, https://doi.org/10.1016/j.heliyon.2023.e15909
Add and “phenolic compound, antioxidant activity”
As the study is mainly focused on phytochemicals and minerals. How it can help in food processing justify the argument in discussion section.
Section 3 started from an extensive discussion or background. I would recommend to start from results. Justify every separate section through discussion and background.
Add future perspective in the conclusion
Moderate English editing is required mostly adjectives are missing in the sentences also some typos must be addressed.
Author Response
Thank You for the constructive and beneficial comments. The all suggestions were included in the article. Please, find our answers to your comments in the table below.
|
The abstract is not informative. Methods description and results are not well presented in the abstract section. Present main findings of the study in the abstract. |
We’ve added the methods, results and main findings of the study in the abstract. |
|
Line 44-48 should be cited with recent studies. It lacks references. The following studies could be cited. https://doi.org/10.3390/molecules28083403, https://doi.org/10.1016/j.heliyon.2023.e15909 |
We’ve added some new recent studies. |
|
Add and “phenolic compound, antioxidant activity” |
We’ve added it. |
|
As the study is mainly focused on phytochemicals and minerals. How it can help in food processing justify the argument in discussion section. |
We’ve justified it. |
|
Section 3 started from an extensive discussion or background. I would recommend to start from results. Justify every separate section through discussion and background. |
We’ve corrected it. |
|
Add future perspective in the conclusion.
|
We’ve added the future perspective in the conclusion. |
|
Moderate English editing is required mostly adjectives are missing in the sentences also some typos must be addressed. |
The revision in all aspects was done in the article, with the consultation of native English speakers and all co-authors. |
Reviewer 3 Report
The manuscript " Biochemical and antioxidant profiling of raspberry plant parts for sustainable processing " is devoted to study of chemical composition of various parts of raspberry (fruits, stems, leaves, flowers, seeds, roots). Mineral composition, antioxidant activity were obtained. Phenolic components were quantified using HPLC. The data obtained in the article are useful for the development of food technologies, waste-free technologies, and biochemistry. In general, works on comparing the content of nutrients in different parts of plants are needed to expand the resource potential. This creates a scientific basis for further search for ways to use all parts of plants.
The manuscript is written in a good language, has a good structure and a clear representation of the data. But there are some shortcomings in the text, especially in the methods description. I think, this manuscript can be published in the Plants journal after minor revision after taking into account general recommendations and comments given below:
1. Section 2.1.: Phenological phase should be specified (i.e. in the BBCH system).
2. Section 2.2. should be named “Reagents” or “Materials”.
3. Section 2.3.: ultrasound parameters should be specified. Also, why did you use these particular extraction conditions?
4. Section 2.4.: if you mentioned the authors of the methods, it would be better to add references.
5. Line 147: how did you obtain 99% ethanol and why did you use it?
6. Section 2.9.: what test was used to determine a statistically significant difference?
7. Lines 230-231: The sentence “The antioxidant activity of raspberry plant…” looks like from another section.
8. All tables: the values and errors should have the same number of decimals.
Author Response
Thank You for the constructive and beneficial comments. The all suggestions were included in the article. Please, find our answers to your comments below.
- Section 2.1.: Phenological phase should be specified (i.e. in the BBCH system).
We’ve specified the phenological phase.
- Section 2.2. should be named “Reagents” or “Materials”.
We’ve named the section.
- Section 2.3.: ultrasound parameters should be specified. Also, why did you use these particular extraction conditions?
We’ve added ultrasound parameters.
- Section 2.4.: if you mentioned the authors of the methods, it would be better to add references.
We’ve added the authors to the references.
- Line 147: how did you obtain 99% ethanol and why did you use it?
It was a technical mistake. We’ve corrected it.
- Section 2.9.: what test was used to determine a statistically significant difference?
We’ve added the test name.
- Lines 230-231: The sentence “The antioxidant activity of raspberry plant…” looks like from another section.
We’ve corrected it.
- All tables: the values and errors should have the same number of decimals.
We’ve corrected it.